# Energy and Macronutrients Intake in Indoor Sport Team Athletes: Systematic Review

**DOI:** 10.3390/nu14224755

**Published:** 2022-11-10

**Authors:** Mónica Castillo, Mar Lozano-Casanova, Isabel Sospedra, Aurora Norte, Ana Gutiérrez-Hervás, José Miguel Martínez-Sanz

**Affiliations:** 1Nursing Department, Research Group on Food and Nutrition (ALINUT), Faculty of Health Sciences, University of Alicante, 03690 Alicante, Spain; 2Ibero-American Network of Researchers in Applied Anthropometry, 04120 Almeria, Spain

**Keywords:** sports nutrition, indoor sport team, energy, macronutrients, nutritional recommendations

## Abstract

Indoor team sports are characterized by matches that are shorter in duration and with frequent substitution (high-intensity intermittent). The main goal of teams is to best cover athletes’ physiological demands, while meeting their dietary intake needs is critical. The aim of this study is to conduct a systematic review of the dietary intake of indoor team sports athletes and to analyze whether they comply with nutritional recommendations. A search of PubMed-MEDLINE, Web of Science, and Scopus databases from 2008 to July 2022 was conducted and 2727 documents were identified. The studies focused on adult professional or semi-professional volleyball, basketball, handball, or futsal athletes. Two independent researchers screened and extracted the data, with 20 documents included after they met the inclusion criteria. Most of the athletes, both men and women, did not meet the official recommendations based on under-consumption (energy and carbohydrates) or over-consumption (fats). In relation to protein, 28.6% of studies met the recommendations, with 50% of those who did not meet it being due to under-consumption. Although there are references on athletes’ dietary intakes, there are no references considering sexes or types of sport. More adapted recommendations are needed in order to more precisely evaluate athletes’ intake to know if they meet their real nutritional requirements.

## 1. Introduction

Team sports can be subdivided according to the field of play into field sports or track and field sports [1]. Court sports utilize indoor facilities and are characterized by smaller playing arenas than field sports, shorter duration matches, and frequent substitutions [2]. Furthermore, team sports can be classified according to whether they use implements such as sticks in hockey or lacrosse, or no implements at all as in volleyball, futsal, football, or rugby, with the participants called team-based ball athletes [3]. These sports are also called high-intensity intermittent sports (HIIS), stop and go sports, or open and closed skills sports due to their physiological and psychological demands [4,5,6].

In both team and individual sports, the players’ goal is the same: to be better and to beat their opponent. To achieve success, it is necessary to measure, evaluate, and have control of certain parameters related to the athlete’s performance, such as the physiological demands of the specific sport [7], sex [8], position within the team [9,10], body composition [8,11], and technical skills, as well as the athlete’s dietary intake [12].

The athlete’s dietary intake is closely related to the physiological demands of their particular sport [13]. The playing actions performed by team sport players follow a pattern of mostly interval–fractional, sub-maximal, and maximal intensity efforts, with interspersed pauses of active and incomplete recovery [14] which are combined with quick decision-making and skills such as agility and speed [15,16,17]. Physiological measurements performed in this pattern of activity show that the demands of this type of sport require a high aerobic capacity and a high glycolytic capacity, as well as a well-developed phosphocreatine breakdown/resynthesis system to recover muscle functions in a short period of time [18,19].

These physiological demands require specific nutritional planning to meet the needs of each athlete according to their age, sex, position, size, etc., or those changes in body composition that are necessary to achieve the best possible performance [20]. Various international associations and scientists in the field of sports nutrition have established general nutritional guidelines and recommendations for team sports, with the intention of ensuring a minimum intake of macronutrients for successful sports performance [4,13,20,21,22].

Macronutrient and energy intake for team sports athletes are defined by associations such as the British Association of Sport and Exercise Sciences (BASES), the International Society of Sports Nutrition (ISSN) [23], the International Olympic Committee (IOC) [12], the Academy of Nutrition and Dietetics, and the American College of Sports Medicine (ACSM) [24], and authors such as Mujika and Burke (2010) [21] and Holway and Spriet (2011) [2], among others. The majority of studies conclude that carbohydrates are the preferred macronutrient in the dietary intake distribution of team sport athletes because of their need due to high-intensity actions [25,26,27].

It is important that athletes meet their energy, macronutrient, and micronutrient needs, as muscle mass must remain stable or increase if the body composition of an athlete needs to be changed. If imbalances exist between energy expenditure and intake, this can lead to a loss of muscle mass, decreased performance, or an increased risk of injury [28]. Different authors have found that athletes competing in team sports, overall, generally do not comply with general dietary recommendations [2,29].

Among team sports, there is a great variety of participation, with football (soccer) being the most played worldwide [30]. That is why there is a large scientific literature on the nutritional intake of footballers, but this is not the case in other team sports such as court sports. Other court team sports can be considered as minority sports because of the relatively low number of people who participate in them worldwide, or because of their lower media visibility [31]. However, it is worth mentioning that they are elite sports, and there are important international competitions such as world championships [32,33] and participation in the Olympic Games for sports such as handball, basketball, and volleyball [34].

The aim of this research is to review the dietary intake of court team sports athletes in the scientific literature, and to analyze whether they comply with the dietary recommendations established by both institutions and expert authors in the field of sports nutrition in team sports.

## 2. Materials and Methods

### 2.1. Design

The present systematic bibliographic review is based on existing evidence on the energy and macronutrients intake of professional or semi-professional volleyball, basketball, handball, or futsal athletes. It was conducted in line with the Preferred Reporting Items for Systematic Review and Meta-Analyses (PRISMA) guidelines [35].

### 2.2. Information Sources and Search Strategy

The databases searched to obtain the most current data were PubMed-MEDLINE (Medlars International Literature Online), Web of Science (WOS), and Scopus. To find the largest number of available articles related to the research aim, the words used in the search strategy were established considering: (1) Team sport; (2) Energy and macronutrient intake; (3) the descriptors of the Medical Subjects Headings (MeSH); (4) other terms described in MeSH as “entry terms”, which include the terminology prior to the setting-up of the MeSH register; and (5) the terms [tiab] or [Title/Abstract] attached to the “entry terms” or MeSH, which allow the localization of these terms in the title and abstract of the articles. The search strategy is shown in Table 1. The timeframe for the search included studies from 2008 until July 2022, to offer the most recent and updated information according to scientific advances in the sports nutrition field, because the nutritional recommendations for athletes have undergone significant changes in the past 14 years [13,26]. Other filters were age (>18 years old), competition level (professional or semi-professional), and studies written in English, Spanish, or Italian.

### 2.3. Eligibility Criteria

The Participants, Intervention, Comparison, and Outcome (PICO) criteria for inclusion and exclusion criteria are shown on Table 2. No limits were placed in relation to the publication status of the study (pre-print, post-print, first online, or final).

### 2.4. Article Management Process

All the documents found were incorporated into the Zotero citation manager in a separate folder, depending on the database where they were found. A common folder was created to detect and delete duplicated articles using the software’s degree of data overlap. The final database was exported in RIS format to be imported into the article screening system for further processing by the researchers.

### 2.5. Study Selection

All retrieved articles were screened in duplicate. The first screening, based on the title and abstract, was independently conducted in all the studies by two authors (ML-C, MC). During the processes of identifying and screening, a third researcher was consulted (JMM-S) to determine if the documents that led to discrepancies between authors had to be included or excluded. The articles eligible for a full text review were then screened by the same authors (ML-C, MC), independently and in duplicate. The rejected articles were then duly identified using the eligibility criterion previously established. Additional reviewers (IS, JMM-S) provided advice when feedback about doubtful documents was required.

### 2.6. Data Extraction

The studies’ characteristics, and the energy and macronutrients intake data, were extracted from all the studies following a blinded and duplicated protocol by two authors (ML-C, MC) using a previously piloted data extraction survey created for this review. The data extraction protocol for this study consisted of the following variables:Study: authors and year of publication.Objective: purpose for which the study was conducted.Study nature/design: cross-sectional, longitudinal, or randomized control trial.Sample: number of subjects by sport and sex.Country/ethnic: geographical area where the data originates from.Sport level: competitive category (professional or semi-professional).Team sports discipline: volleyball, basketball, futsal, or handball.Energy intake: kcal consumed expressed in Kcal/day and Kcal/kg/day.Macronutrients intake: carbohydrate and protein consumed, expressed in g/day and g/kg/day; fat (lipids) consumed, expressed in g/day and % diet.

### 2.7. Study Quality and Data Collection

Two researchers (ML-C, MC) examined the quality of the studies using Strengthening the Reporting of Observational Studies in Epidemiology (STROBE) [37] when studies were of a cross-sectional or longitudinal design, and the Consolidated Standards of Reporting Trials (CONSORT) [38] check-lists when studies were of a randomized controlled trial (RCT) design. A third reviewer (JMM-S) was consulted to solve discrepancies.

## 3. Results

A total of 2727 documents were initially found (366 in MEDLINE, 1665 in Scopus, and 696 in WOS). Figure 1 illustrates the article selection process. After the evaluation of the titles and abstracts, 49 documents were retained for full-text assessment. After the full text was screened, 20 documents were finally included. The characteristics of the studies included are shown in Table 3. The quality of the 20 documents reviewed is shown in Table 4.

### 3.1. Study Characteristics

As Table 3 shows, the majority of the studies included in this systematic review used a cross-sectional design [22,39,40,41,42,43,44,45,46,47,48], while eight were longitudinal [49,50,51,52,53,54,55,56] and just one was a randomized control trial (RCT) [57]. Generally, the aim of the cross-sectional studies was to assess the athletes’ dietary intake, while those using a longitudinal design presented more ambitious objectives such as analyzing associations or evaluations of different interventions in order to improve athletes’ performances.

Of the twenty studies included, five articles reported information about athletes from different sports [22,47,48,55,56]. However, the majority included just one kind of sport: ten documents focused on volleyball [39,40,41,42,43,44,49,50,51,52], with only women in the sample in seven of them [39,40,49,50,51,52], and five studied basketball teams [45,46,53,54,57].

Regarding the sample, 50% of the studies were conducted mainly in the European population [22,39,40,41,45,46,48,49,50,55]. When the studies reported the ethnicity, athletes were predominantly Caucasian [54,56], with only one case studying African Americans [57]. Asian countries were represented in three documents [42,44,47].

In all the studies included in the present systematic review, it was commonly found that they used a small sample, although they included different sports, as they principally focused on a unique team or national teams [22,39,40,42,43,44,45,46,47,48,49,50,51,52,53,54,55,56,57]. This tendency was present independently of the athletes’ professional level (professional or semi-professional).

**Table 3 nutrients-14-04755-t003:** Study descriptive characteristics from the documents included in this systematic review.

Authors and Year	Objective	Study Nature/Design	Sample	Country/Ethnic	Sport Level
Volleyball					
Mielgo-Ayuso et al., 2017 [49]	- To examine relationships between total energy and macronutrient intake in conjunction with controlled training on chronic anabolic/catabolic hormone changes in elite female volleyball players during a 29-week season.	Longitudinal	*n* = 22 female	Spain (*n* = 10), Argentina (*n* = 1), Brazil (*n* = 3), Serbia (*n* = 1)	Professional
Zapolska et al., 2014 [39]	- To assess nutrition, supplementation, and body composition parameters in professional female volleyball players.	Cross-sectional	*n* = 17 female	Poland	Professional
Mielgo-Ayuso et al., 2015 [40]	- To investigate the specific dietary intake of elite female volleyball players. - To assess the impact of specific energy intake in conjunction with periodized training on strength and body composition in female volleyball players over the first 11 weeks of the competitive season. - To compare total energy intake and macronutrients distribution with the established but non-sex-specific recommendations.	Cross-sectional	*n* = 22 female	Spain (*n* = 7), Argentina (*n* = 1), Brazil (*n* = 3), Serbia (*n* = 1)	Professional
Mielgo-Ayuso et al., 2013 [50]	- To evaluate potential changes in the lipid profile that might be induced by 11 weeks of training in female volleyball players. -To collect baseline data on nutrient intake, in order to advise female volleyball players from the Spanish Super League concerning the fat content and quality of their diet during this period.	Longitudinal	*n* = 22 female	Spain	Professional
Mielgo-Ayuso et al., 2013 [41]	- To assess and know the caloric and macronutrient intake by professional volleyball players of the Spanish Superliga for 16 weeks of training or the competition phase and compare with the references marked for the athlete population.	Cross-sectional	*n* = 10 female	Spain	Professional
Valliant et al., 2012 [51]	- To conduct a valuation of dietary intake, nutrition knowledge, and whether education improves dietary intake of collegiate female volleyball players.	Longitudinal	*n* = 11 female (13 initially but 2 were excluded)	USA	Professional
Anderson, 2010 [52]	- To examine the effects of feedback on dietary intake and body composition of female college volleyball players over the competitive season.	Longitudinal	*n* = 15 female in 1st year, *n* = 15 in 2nd year, but *n* = 8 took part in both	USA	Semi-professional
Gamage et al., 2014 [42]	- To examine the dietary intake of Sri Lankan national-level volleyball players during one day of a major competition and evaluate the adequacy of nutrient content and dietary practices compared to current nutrition recommendations for athletes.	Cross-sectional	*n* = 76 (*n* = 43 male *n* = 33 female)	Sri Lanka	Professional
Sesbreno et al., 2021 [43]	- To compare reported dietary intake with nutrition recommendations for sport and health. - To determine the potential association of cognitive restraint on physique traits and emotional eating on knee health. - To assess the association of cognitive restraint and emotional eating behaviors on dietary intake in elite volleyball male athletes.	Cross-sectional	*n* = 22 male, but *n* = 18 completed dietary intake variables	Canada	Professional
Yerzhanova et al., 2018 [44]	- To evaluate and compare the actual nutrition of athletes’ high skills of various sports and the provision of necessary nutrients.	Cross-sectional	*n* = 45 male athletes (*n* = 15 volleyball, *n* = 15 triathlon, *n* = 15 judo)	Kazakhstan	Professional
Basketball					
Zanders et al. 2018 [53]	- To evaluate energy balance and determine the extent to which daily energy needs change across an entire women’s collegiate basketball season.	Longitudinal	*n* = 13 female	USA	Semi-professional
Nepocatych et al., 2017 [56]	- To examine the dietary intake of essential macro and micronutrients from food sources over a 3-day period, and changes in body composition as well as the use of dietary supplements in NCAA Division I female athletes at the beginning compared to after the end of the competitive season.	Longitudinal	*n* = 24 female, but *n* = 20 included (*n* = 10 basketball, *n* = 10 softball)	*n* = 14 Caucasian, *n* = 5 African American, *n* = 1 Hispanic	Professional
Hew-Butler et al., 2022 [57]	- To investigate the effects of modest vitamin D3 supplementation on bone and body composition changes in collegiate basketball players during 3 months of organized summer strength training. - To assess potential body composition differences between African American and Caucasian players. - To investigate performance and other factors which potentially influence total body bone mass.	Randomized Control Trial (RCT)	*n* = 23 (*n* = 8 females, *n* = 10 males), but did not complete post-testing	*n* = 10 African American (*n* = 4 female), *n* = 8 Caucasian (*n* = 4 female)	Semi-professional
Bescós García et al., 2011 [54]	- To investigate vitamin D status in professional male basketball players after wintertime. - To analyze the relationship between serum vitamin D levels and the dietary intake of calcium and vitamin D.	Longitudinal	*n* = 21 (*n* = 13 in 1st season, *n* = 3 in 2nd season, and *n* = 5 in both seasons)	*n* = 16 White Caucasian, *n* = 5 African American.	Professional
Gacek, 2022 [45]	To quantify diet depending on the level of generalized sense of self-efficacy in a group of elite Polish basketball players.	Cross-sectional	*n* = 48 male	Poland	Professional
Tsoufi et al., 2017 [46]	To assess dietary intake and diet quality during training and competition days in a sample of elite basketball players receiving nutritional counseling tailored to the sport’s requirements.	Cross-sectional	*n* = 15 male	Greece	Professional
Handball					
Molina-López et al., 2013 [55]	To evaluate professional handball players’ responses to a nutrition education program in terms of their clinical and nutritional status.	Longitudinal	*n* = 14 male	Spain	Semi- Professional
Volleyball and Basketball					
Papadopoulou et al., 2008 [22]	To record nutritional intake of elite volleyball and basketball female athletes. -To identify differences in nutritional intake between the two sports.	Cross-sectional	*n* = 30 female (*n* = 14 volleyball, *n* = 16 basketball)	Greece	Professional
Volleyball, Basketball, and Handball					
Ahmadi et al., 2010 [47]	To determine the iron status in a group of semi-professional female athletes participating in team ball-sports in Shiraz, Iran.	Cross-sectional	*n* = 45, but *n* = 42 female included *n* = 16 volleyball, *n* = 16 basketball, *n* = 10 handball	Iran	Semi-professional
Aguiar Santos et al., 2011 [48]	To understand the impact of magnesium intake on strength in a pre-season training period in a sample of elite male athletes.	Cross-sectional	*n* = 26 male (*n* = 8 volleyball, *n* = 11 basketball, *n* = 7 handball)	Portugal	Professional

**Table 4 nutrients-14-04755-t004:** STROBE or CONSORT check-list of all studies included.

STROBE																				CONSORT	
	Mielgo-Ayuso (2017) [49]	Zapolska (2014) [39]	Mielgo-Ayuso (2015) [40]	Mielgo-Ayuso, Urdampilleta (2013) [50]	Mielgo-Ayuso, Collado (2013) [41]	Valliant (2012) [51]	Anderson (2010) [52]	Gamage (2014) [42]	Sesbreno (2021) [43]	Yerzhanova (2018) [44]	Zanders (2018) [53]	Nepocatych (2017) [56]	Bescos (2011) [54]	Gacek (2022) [45]	Tsoufi (2017) [46]	Molina-Lopez (2013) [55]	Papadopoulou (2008) [22]	Ahmadi (2010) [47]	Aguiar-Santos (2011) [48]		Hew-Butler (2020) RCT [57]
1a									*							*		*			*
1b	*	*	*	*	*			*	*	*		*	*	*	*	*		*	*	1a	*
2	*	*	*	*	*	*		*	*	*	*	*	*	*	*	*	*	*	*	1b	*
3	*		*										*				*			2a	
4	*	*	*	*	*	*	*		*	*	*	*	*	*	*	*	*	*	*	2b	*
5	*		*	*	*	*		*		*	*		*		*	*	*	*		3a	
6a	*		*	*		*		*	*	*	*		*	*				*		3b	
6b																				4a	
7	*		*			*	*			*	*		*							4b	*
8	*	*	*	*	*	*	*	*	*	*	*	*	*	*		*	*	*	*	5	*
9			*		*	*	*		*	*	*	*	*			*				6a	
10																				6b	
11	*		*						*			*	*		*	*		*		7a	
12a			*	*		*			*	*	*							*		7b	*
12b	*		*					*			*			*						8a	*
12c																				8b	
12d																				9	
12e											*									10	*
13a	*	*	*	*		*	*	*	*		*	*	*					*	*	11a	
13b	*		*	*		*			*		*							*		11b	
13c																				12a	*
14a	*	*	*	*	*	*	*	*	*	*	*	*	*	*		*		*	*	12b	
14b		*			*	*			*								*			13a	*
14c							*		*											13b	
15	*	*	*	*	*	*	*	*	*	*	*	*	*	*		*	*	*	*	14a	
16a	*	*	*	*	*	*	*	*	*	*	*	*	*	*		*	*	*	*	14b	*
16b									*											15	
16c																				16	*
17											*									17a	
18	*	*	*	*	*	*	*	*	*	*	*	*	*	*	*	*	*	*	*	17b	
19	*		*		*	*			*		*	*	*					*		18	
20	*	*	*	*	*	*	*	*	*	*	*	*	*	*		*	*	*	*	19	*
21		*	*	*	*	*		*	*	*	*	*	*	*	*	*	*	*	*	20	*
22							*		*					*		*			*	21	*
																				22	*
																				23	*
																				24	*
																				25	

* meets the STROBE or CONSORT criteria.

### 3.2. Energy and Macronutrients Athletes’ Intake

The information about the athletes’ intake is reported in Table 5. It can be observed that the majority of the studies included reported data about energy and all macronutrients consumed by the athletes [22,39,40,41,42,43,44,45,46,48,49,51,52,53,55,56]. Only three articles included energy intake [47,54,57], while one case studied energy and fat [50].

Focusing on the quantity of each intake, the female athletes’ minimum energy intake was found to be close to 1200 kcal/day [22], with the maximum being 2900 kcal/day [40,41,49,50,52]. However, in men, the minimum was 2000 kcal/day [42,45], while the maximum exceeded 4000 kcal/day [54].

Regarding macronutrients, 1.8 g/kg/day of carbohydrates [22] was the minimum reported for women, while the maximum was 5.7 g/kg/day [52]. In the male sample, the respective values were 3.5 g/kg day [43] and 7.6 g/kg/day [46]. In the case of protein, the women’s minimum and maximum were 0.83 g/kg/day and 2.0 g/kg/day [40,41,49], respectively, whereas men reported values of 1.54 g/kg/day [55] and 2.6 g/kg/day [46]. Finally, the lowest percentage of fat ingested by females was 17.2% [42] and the highest was 39% [56]. Similar results were described for men, with 16.1% being the lowest [42] and 35.7% the highest values [55].

### 3.3. Studies’ Degree of Macronutrient Intake Recommendations Achievement by the Athletes

Table 6 and Table 7 are a compilation of the degree of the official intake recommendations achievement by the different teams/groups of athletes, depending on the sport played, from the studies included.

After a comparison between the studies that reported intake values in the same units as presented in the recommendations, just one study achieved the energy intake recommendation [44], with the other studies reporting lower levels [22,40,41,42,43,46,49,52,53,55,56]. A similar result was found for the consumption of carbohydrates, which was only accomplished by two studies [46,53], while the rest under-consumed [22,40,41,42,43,46,49,52,53,55,56]. For proteins, although ten teams/group of athletes did not achieve the recommendations, in five of them, the athletes consumed more than recommended [40,41,44,46,49], while in the other five, less was consumed than recommended [22,42,51,52]. Contrary to all of these results, in 92.30% of the cases, the athletes’ fat intake was higher than the recommendations [22,40,41,43,44,49,50,51,52,55,56].

## 4. Discussion

The present review comprises research conducted from 2008 to date, regarding the volleyball, basketball, futsal, and handball; adult, female or male; professional or semi-professional athletes’ energy and macronutrients (carbohydrates, proteins, and fats) intake. Several studies relating to the sports searched have been found except for futsal. Overall, by disaggregating the results of the 20 studies included in this review, it was possible to observe a trend regarding energy and carbohydrates, and fats. In the vast majority of studies, the athletes presented an under-consumption of energy and carbohydrates in comparison to the current nutritional recommendations. On the contrary, practically all the sports teams included in the studies had a percentage of fat intake exceeding the official recommendations. In the case of protein, most of the sample met the consumption recommendations of this macronutrient. In addition, in those teams where it was not accomplished, in 50% of cases it was due to under-consumption, whereas it was due to over-consumption in the other 50%.

Of the 20 studies analyzed, 8 used a longitudinal design [49,50,51,52,53,54,55,56], and applied a nutrition education intervention or included nutritional advice to improve a biochemical parameter [49,54], improved the players’ knowledge of nutrition and its implication on sports performance and body composition [50,51,52,53,56], or directly improved the dietary intake of the athletes [22]. In all of the studies, female athletes improved their dietary intake parameters compared to baseline data, although in many cases, these were still lower than the recommendations [13,28,58,59]. Other studies observed the impact of nutrition education in team sports, with most results showing improvements in dietary intake in terms of compliance with the recommendations [60], so this experimental design could be suitable for personalizing the recommendations to each sport modality and to assess their implementation.

Most of the studies analyzed were characterized by a small sample, between 10 and 22 athletes, i.e., one or two complete teams. The exceptions were two studies that utilized larger samples, perhaps because they were samples from smaller geographical areas than the rest, such as the players studied by Gamage in Sri Lanka [42] (*n* = 76) and those studied by Gacek in Poland (*n* = 48) [45]. Studies with larger samples are usually related to more popular elite sports or sports practiced by a much larger number of elite athletes, such as football [61]. Other authors of systematic reviews analyzed studies relating to the effects of nutrition education interventions in professional team sports, with the sample sizes of these studies being similar in number and the subjects generally being complete teams comprised of 10–15 players [62]. Despite the aim of the studies related to elite athletes, it was very common to find small sample sizes [63,64]. This fact was noted by different authors as a problem for extrapolating results and for statistical analyses, but they concluded that with a small sample size within the small group of elite athletes, an adequate power could be obtained [65,66].

### 4.1. Energy Intake

In 2018, the International Olympic Committee (IOC) published the consensus statement on Relative Energy Deficiency in Sport (RED-S), in which the Low Energy Availability (LEA) was defined as “a mismatch between an athlete’s energy intake (diet) and the energy expended in exercise, leaving inadequate energy to support the functions required by the body to maintain optimal health and performance” [12]. It is well known that energy intake is a key factor in sports performance, with both males and females at risk if their diets fall short of their energy requirements [67]. RED-S is less common among males, but above the ideal energy level, it is found in athletes who need to have a very low weight to be able to have more options for better results, including cyclists, rowers, runners, jockeys, and athletes in weight-class combat sports [12]. In addition to the consequences on sporting performance (illness, injury, iron deficiency, impaired cognition, and mood) [67,68], LEA has an impact on the health of athletes, especially in the case of women, as it causes a disruption of the hypothalamic–pituitary–gonadal axis, alterations in thyroid function, changes in appetite-regulating hormones (e.g., decreased leptin and oxytocin, and increased ghrelin, peptide YY, and adiponectin), decreases in insulin and insulin-like growth factor 1 (IGF-1), increased growth hormone (GH) resistance, elevations in cortisol, and interruption of reproductive hormones generation, and consequently the production of a functional hypothalamic amenorrhea (FHA) [67,69,70]. The optimal Energy Availability (EA) established for healthy physiological function in women is typically achieved at 45 kcal/kg Fat Free Mass (FFM)/day (188 kJ/kg FFM/day), with those at risk of LEA being below 30 kcal/kg FFM/day (125 kJ/kg FFM/day) [71].

In those studies analyzed, the measurement value of kcal/kg/day is used instead of kg/FFM/day, but it can be observed that the majority of athletes were between 30–40 kcal/kg/day, with 93.3% below 45 g/kg/day. At some points after educational intervention, 50 g/kg/day was recorded in male athletes [44], but it is notable that in five studies, values at risk of LEA were found with energy intakes below 30 g/kg/day [22,42,46,51,56]. Our findings are in agreement with other studies that reported athletes with LEA or low energy intake with respect to a low carbohydrate intake [21,67].

### 4.2. Intake of Macronutrients

Adaptation to exercise by the athlete is determined by training factors such as the duration, intensity, and type of exercise, as well as exercise frequency, but also by the quality and quantity of pre- and post-exercise nutrition [72]. It is becoming increasingly clear that nutrition can amplify or diminish physiological adaptations initiated by training [73].

#### 4.2.1. Intake of Carbohydrates

Carbohydrate intake recommendations for team sports are in the range of 6–10 g/kg, while in the pre-season or during very intense sessions lasting more than 4 h, this can be increased to 12 g/kg [4,59]. These recommendations may differ in other sports, including cycling or triathlon, which do not involve explosive and intermittent efforts that require the activity of the phosphagen pathway for energy. In this type of sport, dietary carbohydrate loading is played with, with carbohydrate loading usually taking place in the diet prior to some types of training to optimize the performance of the athletes’ aerobic energy pathways [1,29,73]. These strategies of decreasing carbohydrate intake in team sports, generally comprising intermittent and high-intensity efforts, can be detrimental to the athlete’s performance [74], as they compromise energy acquisition through the phosphagen energy pathway, which needs glucose to be able to work normally, thus producing a drop in performance in sports with fast and explosive needs [6,75]. The majority of athletes analyzed in this review (85.7%) were below the recommended low range (6 g/kg). If this reduction in carbohydrates is part of a periodization strategy, with weight-loss objectives, metabolic adaptation, etc., that are controlled with well-programmed variables of training and competition, it does not have to compromise performance. However, in this case, it is associated with a low energy consumption and therefore entails a possible decrease in performance and in the athlete’s ability to assimilate the training, although this issue remain unclear [74]. It is widely recognized that glucose availability is a limiting factor in sports performance, both physical and cognitive, as the central nervous system also relies on glucose as an energy source. This glucose can come from dietary sources or from liver and muscle stores, which are limited and rapidly depleted. It is for this reason that intra-exercise glucose availability is crucial for good sports performance, and also very important in the post-exercise period, so that liver and muscle stores are replenished [29,59]. These results are similar to those found by other authors in female athletes practicing intermittent sports [45,76,77], as well as in sports of different modalities such as tennis, weightlifting [78], or cross-fit [79]

It is crucial for sports dietitians to plan individual and accompanied training schedules and physiological targets in order to achieve optimal performance.

#### 4.2.2. Protein Intake

Most team sports matches include between 150 and 400 high-intensity movement patterns, mainly consisting of running, sprinting, jumping, acceleration, deceleration, and changes in direction, and various sport-specific actions such as tackling, dribbling, and throwing [75,80,81]. These actions have an important eccentric component associated with exercise-induced muscle damage (EIMD), an impaired acute inflammatory response, and an impairment of performance for up to 24–72 h [82]. To repair this muscle damage, the process of muscle protein synthesis (synthesis of amino acids in muscle contractile myofibrils) is key. To initiate this process, the intake of high biological quality proteins is necessary, as well as taking into account the amount of protein per meal and the timing and co-ingestion with other nutrients such as carbohydrates (CHO) to improve their absorption and transport [18,83].

Regarding studies analyzed in this review, there is a greater compliance with protein intake, with the general recommendations of protein for athletes, established between 1.2 and 2 g/kg/day [28], than with the other macronutrients. In fact, there are several cases of excessive consumption, which does not produce any benefit, since for intakes above 2.2 g/kg of protein it has been shown that there is no improvement compared to lower intakes [58]. The percentage of studies in this review showing an intake below the recommended intake, and also those showing an intake above the recommended intake, are equal at 35.7%, with only 28.6% of the studies reporting a protein dietary intake in the recommended range for sports athletes. Other studies that investigated protein intake in team sports described a wide variety of results [60,76,84], including a protein intake of up to 3.4 g/kg in rugby players in a hypertrophy training plan, though usually high protein intakes were accompanied by low carbohydrate intakes in many studies analyzed in the present review. Excessive protein intake, coupled with depleted glycogen stores, could cause an increase in ketone and urea concentration, leading to early dehydration and thus decreased muscle recovery [85]. Food nutrition education for the athlete is important, as the consumption of nutritional protein supplements is widespread [86]; while they are useful tools for improving performance and/or body composition, they must be understood within the recommendations to meet the requirements of all nutrients [23].

#### 4.2.3. Fat Intake

In this review, 87.5% of the studies showed that athletes exceeded the established 25–30% fat recommendations of total daily energy. Fat intake recommendations have not been established for the athlete population, so the recommendations for the adult population, based on the percentage of consumption with respect to the total calorie intake of the diet, are used as the reference [59,87].

Fat does not really represent an important source of energy in high-intensity intermittent sports, given that it cannot act as a substrate in the development of explosive efforts [18]. It is possible that for this reason there are a very limited number of studies [88,89] that have investigated high-fat and low-carbohydrate protocols in these type of sports [88,89]. Paoli et al. [88] suggested this type of strategy when the athlete’s objective is a rapid weight reduction, with the treatment time in this case being 30 days, without any detriment to muscle mass.

It would be interesting to know the dietary source of these fats in order to assess the lipid profile (polyunsaturated, monounsaturated, saturated, or trans fats) that athletes consume in their diet, as not all types of fats have the same function in the body [90]. Adequate consumption of omega-3 fatty acids has been found to have beneficial effects on athletes, including reducing muscle soreness and inflammation after competition [91] or reducing muscle atrophy during periods of injury in combination with protein and carbohydrates [92]. However, a high fat intake can be counterproductive to an athlete’s performance, as it can slow gastric emptying and lead to pre-competition gastrointestinal problems, thus reducing the athlete’s performance [59]. For these reasons, knowing the composition of the fats consumed could be important for recommending an increase or decrease in the consumption of different types of fatty acids [24].

### 4.3. Differences According to Sex: Energy Needs and Dietary Recommendations

In adulthood, males and females present different substrate metabolisms. The only time in life when they use similar substrates is before puberty, largely due to the increase in estrogen concentration in females [93]. In elite female athletes, it has been shown that they oxidize more fat than lipids during the same type of aerobic exercise than men, have a lower amino acid consumption, and excrete less urinary nitrogen urea than male athletes [94]. In another study in a non-athlete population, it was observed that women accumulated more fat than men, and had to reduce their caloric intake to lose weight to a greater extent than men [95]. However, intake recommendations in the sports population are differentiated by physical activity level [96] and not by sex, thus assuming that energy and macronutrient requirements are the same for both, when it has been shown that they are not comparable at the physiological and metabolic levels [93]. One possible explanation is the low sample of elite female athletes until relatively recently. As we have observed, there are few studies showing intakes in line with recommendations for both energy and all macronutrients in female athletes, so perhaps it is worth asking whether the recommendations should be adjusted to their actual/real/specific needs? [93,95,97,98]. Mielgo-Ayuso et al. (2015) suggest that consuming less than the established recommendations of CHO was beneficial overall, and at some point an increased CHO intake without more energy expenditure may yield diminishing returns for strength in this population, as female athletes in their study obtained positive changes in bench press strength after planned strength training with a low-carbohydrate diet [40]. Further research should compile sex-specific CHO recommendations to account for the variance in substrate utilization between males and females [40]. Furthermore, based on their research with strength training for women, Volek et al. (2006) suggested the need for a new energy recommendation of 39–44 kcal/(kg-day)^−1^ for women who mostly perform resistance training [99].

### 4.4. Sources of Information

It is interesting to group the organizations and researchers that established specific recommendations for energy, macronutrients, and micronutrients, as there are several scientific papers in this review, and they may differ in some ranges and recommendations.

The studies analyzed in this paper used 10 different references to compare the daily intake of energy and macronutrients of their subjects, with recommendations from associations, guidelines, and sports nutrition research [28,29,59,85,100,101,102,103,104,105]. On the other hand, other reviews that studied dietary intake in the sports population included other recommendations [96,106], although they may have also included some of those mentioned above.

### 4.5. Limitations

The studies in this systematic review included some with small sample sizes, and their results may be difficult to extrapolate to other populations, although in some cases, depending on the power of the sample, they may be valid for the population they studied; however, none of these studies calculated the power of the sample. Most of the study subjects were Caucasian and European, and this could be a limitation if we want to extrapolate the data obtained to elite athletes of different ethnicity. The measurement instruments used in all the studies were dietary surveys, either 3-day or 1-day recall. These types of instruments may lead to underestimated data, and the authors in most cases did not report data on this possible bias. Therefore, the conclusions should be examined with caution.

The present review focused on court team sports, whose participant numbers are smaller than field sports for two reasons: the number of players per team is smaller (12–15 players) because the players required to play are also fewer (5–6 players); the practice of these sports is a minority compared to other team sports such as football, which has been widely studied in the scientific literature. This review highlights the nutritional intake of these minority sports, so that both nutrition professionals and researchers can learn from the studies’ conclusions.

Further research is needed to discern whether the current recommendations are adequate for intermittent sports, as athletes are not complying with them but are performing well in sport. It would also be interesting to establish recommendations in accordance with the real requirements of the athlete differentiated by sex, as the scientific literature shows that energy requirements are different for both sexes, but the dietary intake recommendations do not differentiate between them.

## 5. Conclusions

This systematic review found that a multitude of references on dietary intake for the athlete population exist but are not specific by sex or type of sport, but by type of training and duration. This personalization should be considered so that professional and semi-professional athletes can comply with the established recommendations, as the vast majority do not comply with the energy and carbohydrate recommendations, and exceed the protein and fat recommendations, which could be a dietary pattern that is detrimental to sports performance. It should be noted that the recommendations in the athlete population are not differentiated by sex, in contrast to the general non-athlete population. It is possible that the samples are much larger in the general population, and there may be a sufficient sample investigated to determine this differentiation, which is not the case in the elite athlete population, where the sample size of studies is small. The authors believe that there is a need for future research in the establishment of dietary intake recommendations that are more in line with each sex and type of sport.

## Figures and Tables

**Figure 1 nutrients-14-04755-f001:**
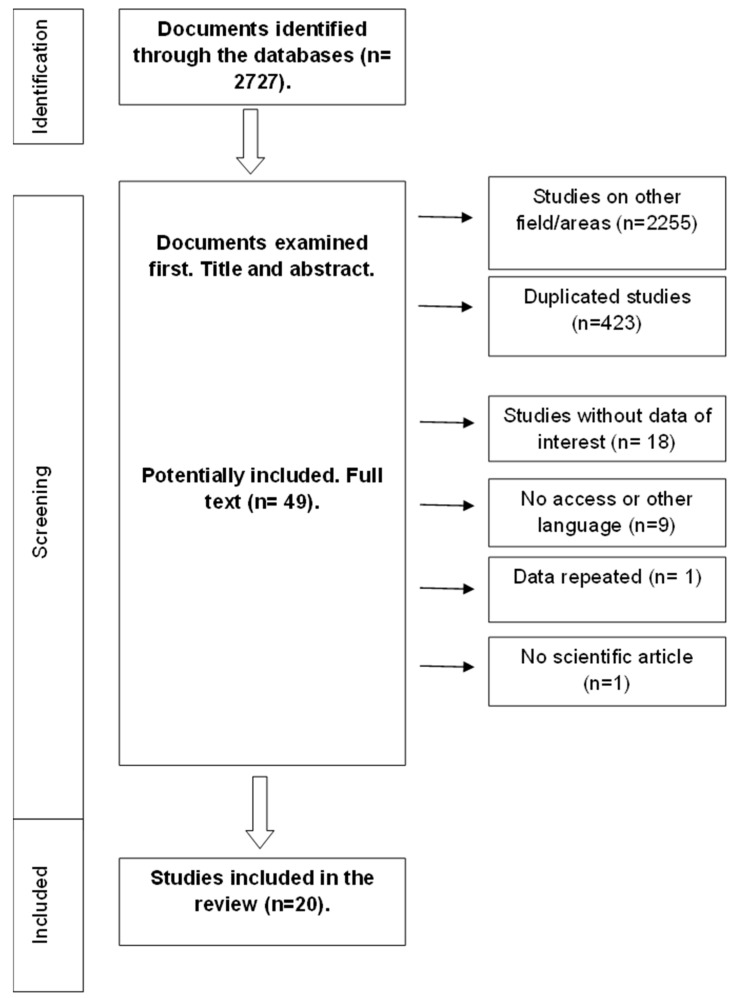
Diagram showing the article selection process.

**Table 1 nutrients-14-04755-t001:** Keywords used for the search strategy on databases.

Concept	Keywords and/or MeSH Terms
Team sport	“Basketball” OR “Volleyball” OR “Basket” OR “Futsal” OR “Five-a-side” OR “Indoor football” OR “Indoor soccer” OR “handball”
Energy and macronutrient intake	“Proteins” OR “Dietary Proteins” OR “Fats” OR “Dietary Fats” OR “Carbohydrates” OR “Dietary Carbohydrates” OR “Energy intake” OR “Macronutrients”
PubMed-MEDLINE search strategy *	
((((((((((((((((“Proteins”[Mesh]) OR “Dietary Fats”[Mesh]) OR “Fats”[Mesh]) OR “Carbohydrates”[Mesh]) OR “Dietary Carbohydrates”[Mesh]) OR “Dietary Proteins”[Mesh]) OR “Energy Intake”[Mesh]) OR (macronutrients[MeSH Terms])) OR (Proteins[Title/Abstract])) OR (Dietary Fats[Title/Abstract])) OR (Fats[Title/Abstract])) OR (Carbohydrates[Title/Abstract])) OR (Dietary Carbohydrates[Title/Abstract])) OR (Dietary Proteins[Title/Abstract])) OR (Energy Intake[Title/Abstract])) OR (macronutrients[Title/Abstract])) AND ((((((((((“Basketball”[Mesh]) OR “Volleyball”[Mesh]) OR (basketball[Title/Abstract])) OR (volleyball[Title/Abstract])) OR (basket[Title/Abstract])) OR (futsal[Title/Abstract])) OR (five-a-side[Title/Abstract])) OR (indoor football[Title/Abstract])) OR (indoor soccer[Title/Abstract])) OR (handball[Title/Abstract]))	

* The search strategy was adapted for each of the databases consulted through the Polyglot Search of the Systematic Review Accelerator tool [36].

**Table 2 nutrients-14-04755-t002:** The inclusion criteria applied in the study followed the Population, Intervention, Comparison, and Outcomes (PICO) strategy.

Population	Intervention	Comparison	Outcomes
Professional or semi-professional athletes from volleyball, basketball, handball, or futsal teams. Sexes: female or male. Healthy athletes Adults.	Dietary intake questionnaires or interviews.	Sexes. Sport. Competition level. Achievement of energy and macronutrients recommendations intake.	Macronutrients (carbohydrates, protein, and fats) and energy intake

**Table 5 nutrients-14-04755-t005:** Sports classification describing energy, carbohydrates, protein, and fats (values ± SD) ingestion from the athletes.

Authors and Year	Energy		Carbohydrates		Protein		Fat	
	Kcal/Day	Kcal/kg/Day	g/Day	g/kg/Day	g/Day	g/kg/Day	g/Day	% Diet
Volleyball								
Mielgo-Ayuso et al., 2017 [49]	Period 1 *: 2890 ± 88 Period 2: 2790 ± 50 Period 3: 2810 ± 75 Mean: 2830 ± 50	42.4 ± 2.0 40.5 ± 2.1 41.8 ± 2.0 41.8 ± 1.9	305± 11 297 ± 6.1 305 ± 9 302 ± 6.5	4.5 ± 0.2 4.4 ± 0.2 4.5 ± 0.2 4.5 ± 0.2	143 ± 6.8 135 ± 6 138 ± 7 139 ± 6.9	2.1 ± 0.1 2.0 ± 0.1 2.0 ± 0.1 2.1 ± 0.1	115 ± 6 106 ± 4 108 ± 3.7 109 ± 3.2	35.5 ± 1.6 34.9 ± 1.2 35.5 ± 1.1 35.3 ± 1.0
Zapolska et al., 2014 [39]	1909.6 ± 560.1		221.5 ± 101		113.5 ± 28		69.9 ± 25.9	
Mielgo-Ayuso et al., 2015 [40]	2835 ± 178	40.7 ± 5.2	301 ± 36	4.3 ± 0.6	140 ± 18	2.1 ± 0.4	114 ± 16	36.1 ± 4.6
Mielgo-Ayuso et al., 2013 [50]	2840 ± 268	41 ± 6					113 ± 20	35.6 ± 4.8
Miego-Ayuso et al., 2013 [41]	2751 ± 176	41.1 ± 6.42	301 ± 21.6	4.47 ± 0.53	135 ± 19.1	2.03 ± 0.43	107 ± 11.9	35.1 ± 3.25
Valliant et al., 2012 [51]	Period 1: 1756 ± 557.5 Period 2: 2178 ± 491.8	24.0 ± 8.6 29.4 ± 7.5	224 ± 64.4 304.0 ± 79.9	3.08 ± 1.1 4.15 ± 1.3	69.3 ± 26.8 84.0 ± 20.5	0.9 ± 0.3 1.14 ± 0.3	67.4 ± 27.8 69.0 ± 24.8	33.7 ± 6.4 27.9 ± 5.2
Anderson, 2010 [52]	Beginning: - Period 1: 2425.7 ± 292.1 - Period 2: 2883.9 ± 195.6 Peak: - Period 1: 2888.9 ± 215.5 - Period 2: 2534.9 ± 248.6 After: - Period 1: 2435.0 ± 190.6 - Period 2: 22,566 ± 270.2	35.3 ± 4.6 40.7 ± 2.2 40.8 ± 2.8 35.4 ± 3.9 34.7 ± 2.7 31.3 ± 3.8	339.5 ± 41.0 402.5 ± 23.1 340.2 ± 34.0 350.5 ± 48.3 315.2 ± 19.3 310.9 ± 37.4	5.1 ± 0.7 5.7 ± 0.3 4.8 ± 0.4 5.1 ± 0.8 4.5 ± 0.3 4.3 ± 0.5	77.5 ± 10.0 106.0 ± 12.3 85.8 ± 12.9 82.2 ± 8.1 79.5 ± 9.0 81.7 ± 9.5	1.1 ± 0.1 1.5 ± 0.1 1.1 ± 0.2 1.1 ± 0.1 1.1 ± 0.1 1.1 ± 0.1	89.1 ± 15.1 98.7 ± 11.5 99.6 ± 10.3 90.6 ± 8.3 78.5 ± 14.8 79.4 ± 11.3	31.7 ± 2.8 30.2 ± 2.3 32.6 ± 1.7 31.6 ± 1.6 34.9 ± 1.5 30.8 ± 1.7
Ahmadi et al., 2010 [47]	2266.00 ± 835.9							
Papadopoulou et al., 2008[22]	1167 ± 130	15.85 ± 1.71	139.01 ± 37.12	1.89 ± 0.48	64.00 ± 14.75	0.87 ± 0.19	41.32 ± 8.18	31.94 ± 5.87
Gamage et al., 2014 [42]	Male: 2309 ± 365 Female: 1829 ± 383 All: 2100 ± 441	30.9 ± 5.7 30.8 ± 6.8 30.8 ± 6.2	420.5 325.8 379.2	5.6 ± 1.0 5.5 ± 1.1 5.6 ± 1.0	64.1 53.9 59.8	0.85 ± 0.17 0.98 ± 0.53 0.91 ± 0.37	41.3 34.9 38.7	16.1 17.2 16.6
Sesbreno et al., 2021 [43]	3034 ± 1345	33 ± 15	325 ± 105	3.5 ± 1.3	161 ± 34	1.7 ± 0.4	119 ± 37	35 ± 7
Yerzhanova et al., 2018 [44]	4033 ± 546.1	51.64	472.04 ± 101.02	6.04	162.20 ± 48.2	2.08	158.97 ± 46.31	35.20 ± 8.12
Basketball								
Zanders et al., 2018 [53]	Period 1: 2506 ± 271 Period 2: 2354 ± 533 Period 3: 2326 ± 456 Period 4: 2571 ± 334 Period 5: 2422 ± 276	33.7 ± 3.1 31.9 ± 7.8 31.5 ± 7.3 33.8 ± 3.7 32.7 ± 4.9	282.4 ± 60.3 272.2 ± 73.2 244.8 ± 42.2 299.9 ± 36.4 263.2 ± 36.8	3.8 ± 0.7 3.7 ± 1.1 3.3 ± 0.7 4.0 ± 0.4 3.6 ± 0.7	97.9 ± 18.8 87.3 ± 13.9 87.5 ± 17.0 78.0 ± 13.9 84.7 ± 16.3	1.31 ± 0.22 1.18 ± 0.19 1.19 ± 0.28 1.05 ± 0.19 1.15 ± 0.26	113.0 ± 26.1 98.4 ± 27.1 112.7 ± 29.3 87.3 ± 18.8 93.3 ± 28.5	
Ahmadi et al., 2010 [47]	1778.90 ± 652.8							
Nepocatych et al., 2017 [56]	Period 1: 2208 ± 373 Period 2: 2567 ± 834	29 ± 8 34 ± 15	254 ± 51 304 ± 74	3.4 ± 1.0 4.1 ± 1.5	92 ± 29 97 ± 38	1.3 ± 0.6 1.4 ± 0.7	87 ± 19 111 ± 42	35 ± 5 39 ± 7
Papadopoulou et al., 2008 [22]	1344 ± 250	19.29 ± 4.41	174.30 ± 49.92	2.51 ± 0.86	58.30 ± 11.15	0.83 ± 0.17	46.18 ± 11.82	31.25 ± 7.27
Hew-Butler et al., 2022 [57]	Vitamin D female group: 2193.0 ± 864.3 Placebo female group: 1806.3 ± 39.9 Vitamin D male group: 2117.8 ± 1128.3 Placebo male group: 1900.2 ± 470.9							
Bescós García et al., 2011[54]	4284 ± 701							
Gacek, 2022 [45]	1798.5 ± 547.9		258.2 ± 105.9		79.3 ± 23.4		58.5 ± 24.5	29.4 ± 9.4
Tsoufi et al., 2017 [46]		Training days: 26.0 (21.7, 26.4) Competition days: 19.5 (19.3, 22.1)		7.6 ± 1.5 6.8 ± 0.9		2.6 ± 0.6 2.2 ± 0.2		
Handball								
Ahmadi et al., 2010 [47]	2136.90 ± 679.6							
Molina-López et al., 2013 [55]	Period 1: 2974.50 ± 211.11 Period 2: 3355.08 ± 325.31 Period 3: 3328.64 ± 306.13	34.45 ± 3.56 38.91 ± 4.15 38.54 ± 2.94	360.91 ± 27.64 421.50 ± 4.88 416.80 ± 4.82	4.17 ± 0.41 4.88 ± 0.60 4.82 ± 0.36	133.43 ± 14.32 146.64 ± 35.64 147.04 ± 25.51	1.54 ± 0.22 1.70 ± 0.44 1.70 ± 0.33	118.57 ± 22.52 132.22 ± 17.75 129.57 ± 21.79	35.71 ± 4.88 35.51 ± 3.81 34.92 ± 4.01
Volleyball, Basketball, and Handball								
Aguiar-Santos et al., 2011 [48]	2654 ± 821		331 ± 123		121 ± 32		88 ± 34	

* Period: number of phases to show athletes’ intakes by time-periods analyzed in original documents.

**Table 6 nutrients-14-04755-t006:** Degree of achievement of the dietary intake recommendations for athletes, depending on the sample from all the studies included in this systematic review [28,58,59].

Authors and Year	Energy (kcal/kg/Day)	Carbohydrate (g/kg/Day)	Protein (g/kg/Day)	Fats (%)
	45–50	6–10	1.2–2	25–30
Volleyball				
Mielgo-Ayuso et al., 2017 [49]	No (↓)	No (↓)	No (↑)	No (↑)
Zapolska et al., 2014 [39]				
Mielgo-Ayuso et al., 2015 [40]	No (↓)	No (↓)	No (↑)	No (↑)
Mielgo-Ayuso et al., 2013 [50]	No (↓)			No (↑)
Miego-Ayuso et al., 2013 [41]	No (↓)	No (↓)	No (↑)	No (↑)
Valliant et al., 2012 [51]	No (↓)	No (↓)	No (↓)	No (↑)
Anderson, 2010 [52]	No (↓)	No (↓)	No (↓)	No (↑)
Ahmadi et al., 2010 [47]				
Papadopoulou et al., 2008 [22]	No (↓)	No (↓)	No (↓)	No (↑)
Gamage et al., 2014 [42]	No (↓)	No (↓)	No (↓)	No (↓)
Sesbreno et al., 2021[43]	No (↓)	No (↓)	Yes	No (↑)
Yerzhanova et al., 2018 [44]	Yes	Yes	No (↑)	No (↑)
Basketball				
Zanders et al., 2018 [53]	No (↓)	No (↓)	Yes	
Ahmadi et al., 2010 [47]				
Nepocatych et al., 2017 [56]	No (↓)	No (↓)	Yes	No (↑)
Papadopoulou et al., 2008 [22]	No (↓)	No (↓)	No (↓)	No (↑)
Hew-Butler et al., 2022 [57]				
Bescós García et al., 2011[54]				
Gacek, 2022 [45]				Yes
Tsoufi et al., 2017 [46]	No (↓)	Yes	No (↑)	
Handball				
Ahmadi et al., 2010 [47]				
Molina-López et al., 2013 [55]	No (↓)	No (↓)	Yes	No (↑)
Volleyball, Basketball, and Handball				
Aguiar-Santos et al., 2011 [48]				

Note: ↓ and ↑ symbols indicate in what sense athletes did not meet the official intake recommendations.

**Table 7 nutrients-14-04755-t007:** Percentage of studies in which athletes are below, within, or above the ranges of dietary energy and macronutrient recommendations.

	Recommendation	Percentage of Compliance According to Recommendation
Energy (Kcal/Kg/day)	<45 45–50 >50	93.3 6.7 0
Carbohydrates (Kcal/Kg/day)	<6 6–10 >10	85.7 14.3 0
Proteins (Kcal/Kg/day)	<1.2 1.2–2 >2	35.7 28.6 35.7
Fat (%)	<25 25–30 >30	7.1 7.1 85.7

Energy intake recommendations: Thomas, D.T.; Erdman, K.A.; Burke, L.M. (2016) [24]; and González Gross, M. (2020) [59]. Macronutrients recommendations: Thomas, D.T.; Erdman, K.A.; Burke, L.M. (2016) [24]; and Kerksick, CM. (2018) [28].

## Data Availability

Not applicable.

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
