# Peer review of "Energy and Macronutrients Intake in Indoor Sport Team Athletes: Systematic Review"

_nutrients, 2022, doi:10.3390/nu14224755_

Round 1

Reviewer 1 Report

The document is well written and is of the utmost importance for coaches and nutritionists, especially for the latter with the aim that they are of the greatest need in high-performance sport.
However, in the methodology and results section there are some inconsistencies that need to be resolved.
General features
1- In the summary the sport of futsal appears but later, in the results there is no work of this sport. And, on the other hand, softball appears in the results (table 3), and then it is never named again.
2- In section 2.7 it is described that STROBE and CONSORT are used to analyze the quality of the studies, but in table 4, it appears that some articles have been analyzed with one and others with another. Then it is required to specify the use of one or the other, as well as if only one person or several people carried out this task.
3- In tables 4, 5 and 6, the number of articles analyzed does not correspond to the results obtained in the systematic review. That is, 20 articles were obtained, and on the other hand, in table 4 there are 19 analyzed, and in table 5 there are 23. In addition, in this last table, the articles analyzed do not follow the same classification as in table 3.
4- Attention to the use of gender and sex interchangeably, since there are indeed differences in the use of this term. Always use the same term when referring to the same concept.
5- Review the way of referencing the journals in the bibliographical references section since they do not always follow the rule of the abbreviated title.
6- The font size in the tables is not always the same.

Specific aspects
They are highlighted in the attached file.

Author Response

Authors’ Reply to the Reviewers – Nutrients- 2010607 “Energy and macronutrients intake in indoor sport team athletes: Systematic review”.

Reviewer 1

The document is well written and is of the utmost importance for coaches and nutritionists, especially for the latter with the aim that they are of the greatest need in high-performance sport.

However, in the methodology and results section there are some inconsistencies that need to be resolved.

General features

  • In the summary the sport of futsal appears but later, in the results there is no work of this sport. And, on the other hand, softball appears in the results (table 3), and then it is never named again.

Response of the authors: The authors used futsal in the search strategy but no study was found on the diet of this type of athlete that met the PICO characteristics, so it does not appear again in the text. This aspect has been explained in discussion section.

Softball sport appeared in an article that was selected because it also studied basketball athletes. However, the softball data were not considered because it did not belong to the sport modalities studied, and consequently, the sport has been deleted from the table. 

  • In section 2.7 it is described that STROBE and CONSORT are used to analyze the quality of the studies, but in table 4, it appears that some articles have been analyzed with one and others with another. Then it is required to specify the use of one or the other, as well as if only one person or several people carried out this task.

Response of the authors: The authors have added information about when they have used one or the other quality control according to the study design. The people who carried out this quality control as well as the supervision of the task are also specified in the text. All this information is in lines 151 to 155.

  • In tables 4, 5 and 6, the number of articles analyzed does not correspond to the results obtained in the systematic review. That is, 20 articles were obtained and, on the other hand, in table 4 there are 19 analyzed, and in table 5 there are 23. In addition, in this last table, the articles analyzed do not follow the same classification as in table 3.

Response of the authors: In the present systematic review, 20 articles were obtained and, according to reviewer’s suggestions one reference missing in table 4 has been added. Moreover, all the tables have been revised and all the articles have been rearranged to follow the same order in the tables except in tables 5 and 6, where articles are classified according to the sports included. Table 3 is about the descriptive characteristics of the studies, that are common for all the sports included in the same study, however, when these articles are classified attending to each sport included (tables 5 and 6) some of them are repeated because the results are differentiated by sport. For this reason, tables 5 and 6 have 23 references and not only 20.

  • Attention to the use of gender and sex interchangeably, since there are indeed differences in the use of this term. Always use the same term when referring to the same concept.

Response of the authors: Thank you for your appreciation. The authors have changed gender by sex throughout the article.

  • Review the way of referencing the journals in the bibliographical references section since they do not always follow the rule of the abbreviated title.

Response of the authors: The authors have revised the bibliography and corrected errors in journal abbreviations.

  • The font size in the tables is not always the same.

Response of the authors: The authors are grateful for the reviewer's comment. The font types and sizes of all tables have been revised.

Specific aspects

They are highlighted in the attached file.

Response of the authors: The authors have revised the specific aspects in the file and all the changes have been accepted.

Reviewer 2 Report

This excellent and outstanding systematic review examines the nutritional intake in (semi-) professional athletes in ball team sports -- with an interesting diference in needs (but not in guidelines) between men and women. Despite an emphasis on association football and caucasian athletes altogether, the study selection and data extraction are well performed, and the article is well written.

I only have one remark: the authors should discuss why a meta-analysis was not performed, to strengthen the article, regardless of the outcome (as the studies are indeed quite heterogeneous in design).

Author Response

Authors’ Reply to the Reviewers – Nutrients- 2010607 “Energy and macronutrients intake in indoor sport team athletes: Systematic review”.

Reviewer 2

This excellent and outstanding systematic review examines the nutritional intake in (semi-) professional athletes in ball team sports -- with an interesting diference in needs (but not in guidelines) between men and women. Despite an emphasis on association football and caucasian athletes altogether, the study selection and data extraction are well performed, and the article is well written.

I only have one remark: the authors should discuss why a meta-analysis was not performed, to strengthen the article, regardless of the outcome (as the studies are indeed quite heterogeneous in design).

Response of the authors: Although authors are not experts in meta-analysis, they consider that conducting a meta-analysis requires certain common characteristics in primary studies such as: the effect size index and homogeneity in terms of methodology, statistics and clinical practice. The effect size index should be equal in all studies but in the case of this review, not all articles presented the data about effect sizes of the samples.

Along the same lines, homogeneity in terms of methodological, statistical and clinical characteristics is exhibited. In this sense, primary studies do not show homogeneity in these aspects.

Round 2

Reviewer 1 Report

Dear authors,

Thank you very much for accepting the proposed changes. Only one minor thing, and that is the table 4 is not well edited, since there are columns that are cut off.